# Hydrocode Investigations of Terminal Astroballistics Problems during the Hypothetical Future Planetary Defense System’s Space Mission

**DOI:** 10.3390/ma15051752

**Published:** 2022-02-25

**Authors:** Maciej Mroczkowski, Stanisław Kachel, Adam Kozakiewicz

**Affiliations:** 1Institute of Optoelectronics, Military University of Technology, 00-908 Warszawa, Poland; maciej.mroczkowski@wat.edu.pl; 2Institute of Aviation Technology, Faculty of Mechatronics, Armament and Aerospace, Military University of Technology, 00-908 Warszawa, Poland; adam.kozakiewicz@wat.edu.pl

**Keywords:** astronautics, astrodynamics, terminal astroballistics, hydrocode, warheads, modeling and simulation, rock material asteroid, network-centric system architecture, space missions, navigation, planetary defense modeling

## Abstract

The article is devoted to the preliminary concept of the Future Planetary Defense System (FPDS) emphasizing astroballistics. This paper is intended to support international efforts to improve the planetary security of Earth. The work covers three areas of knowledge: astronautics, astrodynamics, and astroballistics. The most important part of the presented article is dynamic, contact combat modeling against small, deformable celestial bodies. For these purposes, the original, proprietary hydrocode of the free particle method (HEFPM-G) with gravity was used. The main aim of combat is to redirect potentially hazardous objects (PHOs) to orbits safe for Earth or destroy them. This concept’s first task is to find, prepare, and use dynamic three-dimensional models of the motion of celestial bodies and spacecraft or human-crewed spaceships in the solar system’s relativistic frame. The second task is to prepare the FPDS’ architecture and computer simulation space missions’ initial concepts in the internal part of the solar system. The third and main task covers simulating, using hydrocodes, and selected methods of fighting 100 m diameter rock material asteroids.

## 1. Introduction

In this paper, an original proper hydrocode of free particles method (HEFPM-G) with gravitation [1] was applied to simulate the terminal astroballistics problems with three methods of contact deflection of virtual potential hazardous asteroids (PHAs). PHAs are small celestial bodies in the solar system. The issue of collisions of small celestial bodies has been explored since the early 1950s. Some collisions can generate small solar system bodies (SSSBs), which can become as dangerous as PHAs and eventually hit the Earth. The most interesting from the future defense space system (FPDS) point of view is cosmic collisions in the inner solar system space area. The greatest probability of their occurrence concerns the space close to the ecliptic plane.

The most crucial hydrocode modeling of dynamic contact interactions of rock material asteroids is a part of the computational mechanics of continuous media. It is a modeling of dynamic events taking place in close contact for various deformable materials, which were initially solids in a state of hydrostatic equilibrium. These events include asteroid collisions or a spacecraft hitting an asteroid, an asteroid impact onto Earth, and any other impact on asteroids through high-energy materials or laser radiation. This first modeling class is related to terminal astroballistics.

The events described above take place in the inner solar system. From the point of view of the Future Planetary Defense System, potential threats must be fought as far away from Earth as possible, as soon as possible after their detection and identification, and as close to their origins as possible. One or more spaceships with a crew of astronauts and destroyers (autonomous spacecraft) on rendezvous or fly-by missions must fly to the vicinity of potentially hazardous objects (PHO). The purpose and action of each spacecraft or spacecraft-destroyer during the FPDS mission is assigned by the command and control (C2) center. The processes of navigation in the inner space of the solar system during flights to the vicinity of PHO require many solutions to the fundamental problem of celestial mechanics. It is a complex problem to determine the orbits and 3D trajectories for the maneuverability of spacecraft. Spaceships or spacecraft are treated as material points or even perfectly rigid bodies. It should be mentioned that the inclusion of the gravity of the N-bodies of the solar system is a severe inconvenience, and their ephemeris must be quoted at every critical moment and every important place. This second class of modeling is related to astronautics and astrodynamics.

This paper’s primary goal was to simulate, as realistically as possible, the impulse interactions on small, dangerous space bodies. This is the decisive and final phase of the original project of the FPDS Space Mission. However, the order of issues discussed in this publication results from the imposed chronology of space system preparation primary ideas and space mission implementation rules [2]. From the mission navigator’s point of view, it is necessary to start the navigation in an interesting area of the solar system space, where the beginning of the threat to Earth has been detected. This procedure requires the systematic use of the computational methods of astronautics and astrodynamics to model and simulate motion until the spacecraft’s close encounter with the PHO.

The user should then move to model the processes of terminal ballistics and choosing an effective method to remove the threat.

The main contributions of this paper are as follows. The first part of the article presents an architecture and modus operandi of the FPDS. Preliminary design and development of the FPDS’ space missions including navigation and mission dynamics simulation is prepared using an open-source space mission analysis and design tool (e.g., Asterank and Trajectory Browser or GMAT) are presented in the second part. The third part of the article is devoted to computer hydrocodes (HEFPM-G) and the modeling and simulation of asteroid–asteroid collision, laser radiation effects on an asteroid, and the FPDS spacecraft’s warhead contact interaction with the small celestial body. The authors formulated the main planetary defense problem (MPDP) in this paper. The proposition of this problem solving was realized by preparing the concepts, architecture, and modus operandi of the FPDS mission. Finally, a series of realistic simulations were conducted using hydrocodes to deflect or destroy dangerous asteroids. The summary and conclusions can be found in the fourth part of the article.

The necessary participation of astronauts in FPDS bases on Earth, Moon, and Mars and manned space vehicles introduces the need to model the astronautic problems of human-crewed spaceships in the near future. The FPDS is closely related to asteroid observation and identification programs [3].

### 1.1. Mathematical–Physical Model of Solar System Dynamics in Barycenter Relativistic Frame and Future Planetary Defense System (FPDS) Network-Centric Architecture

The first part of this article provides the basic mathematical–physical models for solar system dynamics in relativistic frames [4,5]. Next, an abridged version of the description of the FPDS network-centric architecture (FPDS is distributed in the inner part of the solar system) is presented [6].

### 1.2. NASA Ames Research Center Trajectory Browser’s Example of FPDS Mission to Potentially Hazardous Object (PHO)

In the second part of the article, the NASA Ames Research Center Trajectory Browser was used. The NASA Trajectory Browser made it possible to obtain results from the classic astrodynamic Lambert problem. The chase trajectory was obtained by solving a typical two-body problem. The problem of navigating to potentially dangerous small celestial asteroids has been resolved. In the simplest case, this makes it possible to effectively navigate an autonomous or manned spacecraft and perform planetary maneuvers such as rendezvous or fly-bys to reach the close vicinity of a dangerous asteroid. Under such conditions, it is possible to obtain spacecraft warhead–asteroid contact interactions in an individual action. In this paper, asteroids with a diameter of 100 m (which are very common, as suggested by Mark Boslough [7]) were selected for analysis because they are a significant risk to the human population. However, there are still problems with recognition, spatial situational awareness, and decision-making simulations in more complex cases generated in the solar system’s deep space [8]. The sources of the problems are asteroid and comet collisions, gravitational interactions, and threats from interstellar intruders.

In the next part, a professional open-source computer system developed by a team composed of NASA, private industry, and a range of academic, public, and private contributors, the General Mission Analysis Tool (GMAT) [9], is shown. The GMAT has the possibility to obtain precise results from simulating astrodynamic problems and navigating to potentially dangerous small celestial bodies. Relationship modeling in the FPDS network-centric architecture [10,11,12] is only partially possible inside the GMAT. This should be conducted using MATLAB’s web tools or software written in Python, adapted to the GMAT interface. Finally, after improvements in the fully network-centric, autonomous, and multi-spacecraft FPDS missions, the technology of asteroid deflection or destruction is ultimately expected to protect Earth from impact. The necessary definitions, an example of a FPDS space mission scenario, and the initial-boundary conditions of selected simulations of spacecraft dynamics and small solar system bodies (SSSB) are also discussed. This section introduces a new class of astronomical objects from a planetary defense perspective: the interstellar object [13].

### 1.3. Hydrocode Modeling and Simulations of Potentially Hazardous Asteroids (PHAs) Deflection or Destruction, and the Main Planetary Defense Problem

The third part presents basic models of the dynamics of solid small celestial bodies collisions in the presence of gravity as well as models of impulse interactions on PHAs with the use of profiled explosives and laser radiation. The purpose of this computer modeling is to find parameters of the spacecraft’s warhead operations that will allow for a solution to the main planetary defense problem (MPDP, see Section 2) either by the deflection or even destruction of a certain class of dangerous asteroids (100 m in diameter) on a collision course with Earth. Destruction of asteroids should be conducted by crushing or spalling and dispersion. The work, at this stage, assumes deflection by explosives or impact or laser interactions, but the collection of available methods will be expanded as the project develops. The problem of the destruction and dispersion of asteroids and comets will be discussed in more detail including nuclear technologies, in subsequent articles. The simulations were performed using updated proper hydrocode of the free particles method with gravitation prepared for solving the shock-wave physics problems. This hydrocode has been comprehensively verified, calibrated, and positively validated during the modeling of many cases of deformable celestial body collisions, as shown in [1] and [14,15,16,17,18,19,20,21]. As spacecraft warhead operations are most effective at close range, the final part of the Future Planetary Defense System (FPDS) mission will start at the end of the precision deep space navigation phase. To maintain the responsiveness of the defense system, three-stage armed space vehicles capable of independent operations are planned (see Section 1.2). Depending on how the situation develops, it will be possible to prepare 3D maneuvers in space flight including fly-bys, rendezvous, or approach a collision straight from the retrograde orbit. Solving the main problem of planetary defense results from gradually building up a proper system. Basic requirements include a high autonomy level with sensors to navigate, recognize, and track, and the warheads’ actuators to intercept and redirect or destroy “almost all” PHOs. PHOs represent all asteroids and comets that have sufficient geometric, material, and dynamical parameters to threaten Earth’s population. The issue of sensor subsystems is closely related to the space metrology problems of the spacecraft atomic and pulsar time. The global synchronization between the spacecraft and local synchronization onboard the spacecraft are very important in the deep space of the solar system. The primary system, applied to the solar system scale, is a set of nodes and connections between them. A proper architectural model of the distributed network-centric planetary defense system, where the nodes are autonomous spacecraft with autonomous physical sensors on board, is proposed. Laser communications connect the nodes. In the presence of the Sun, planets, and small celestial bodies, relativistic theories for a physical description of dynamical phenomena are applied. As recommended by the IAU, a post-Newtonian equation of motion model in a relativistic coordinate system is presented. The proposed accuracy of the measurement sensor reached the level of space quantum metrology tools (see Section 3.2—SI 2018).

## 2. Initial Remarks on Paradigm of Research in Science Using to Solve the Main Planetary Defense Problem (MPDP)—Architecture and the Final Configuration of FPDS Mission Geometry

This article is a development of research into problems in modeling the Future Planetary Defense System (FPDS) [2]. The focus has shifted from the synchronization and coordination (with the correlation) problems of autonomous sensor systems to the modeling of local hydrocode interactions. At an early conceptual stage, it is important to make effective use of computer experiments. In this way, modern or future technologies of highly effective combat systems should be found and quantified. It is known that the best include short-range action or even contact interaction using impulsive [Holsapple] methods to disrupt and disperse PHOs or deflect them. This means that the FPDS architecture (see Section 2.3) that guarantees the possibility of early contact action on a PHO will be of fundamental importance. Architecture is the structure of components, their relationships, and the principles and guidelines governing their design and evolution over time [DoD Architecture Framework Ver.1.0, based on IEEE STD 610.12]. FPDS architecture is an evolution of the C4ISR Architecture Framework toward NCOW, SSA, and ODDA from the operational view (OV), system view (SV), and technical standards view (TV). However, the FPDS, as a whole, is a large system in inner solar system space, which must work in a strictly defined sequence of steps and needs synchronization.

To do this, architecture geometry modeling starts with the relationship between A, B, C, …, and Z clocks synchronization and coordination. Correlations should be understood in the sense of the Poincare–Einstein convention on a closed road. This issue is important for FPDS clocks in orbit around the Sun. However, even in the Earth layer of FPDS, the amount of necessary relativistic corrections for FPDS’ spacecraft clocks can be estimated (e.g., from the global positioning system (GPS) system satellite clocks in orbit around Earth). The next step is modeling the FPDS synchronization events as a coordination of the joint operation of the entire future planetary defense system problem and planetary protection [22].

This paper refers to some relativistic models. (a) In the context of special relativity (SR), in the case of synchronization and correlation and data transmission exchange by using electromagnetic wave propagation and the propagation of laser beams as a photon stream where the latter also applies to the use of a laser to destroy asteroids; and (b) within the framework of approximations of general relativity (GR) in the case of the motion of massive bodies.

In this work, the extended paradigm of research in science was used (Figure 1). It specifies the interdependence between theory, observation, and experiment, which together form the best mathematical and physical models of a computer experiment capable of simulating future events. The paradigm is applied through the first pillar: analysis, selection, and adaptation of the appropriate theory. In the second pillar, the experiments and the observations are conducted. It allows for the verification of the selected theory. In the third pillar, there are computational experiments in the virtual laboratory, which include simulations of physics in virtual reality (VR), and the mathematical, physical, and numerical models are prepared. After writing and running the programs for the simulation and visualization of calculations, it is possible to compare the results from VR with reality.

The sequence of actions presented in the paper with the use of precise computational experiments is the correct conclusion from the scientific paradigm.

### 2.1. The Main Planetary Defense Problem (MPDP) of the Future Planetary Defense System (FPDS)

As above-mentioned, the solving of the real main planetary defense problem (MPDP) means that almost all potentially hazardous objects (PHOs) will be redirected or destroyed. The modeling and simulation in virtual reality of the real dynamical processes in the solar system are very difficult. In this case, very high accuracy is required to guide warheads toward asteroids. Additional problems are connected with the modeling of a proper FPDS with high autonomy level spacecraft and their subsystems to recognize, track, intercept, and redirect or destroy almost all asteroids and comets that have sufficient parameters to threaten Earth’s population. Real MPDP ends with the destruction of the Earth-threatening object. In VR, new measures will have to be developed to assess the degree of destruction of the attacked asteroid. Simulating the processes taking place inside the attacked asteroid is very difficult to model and interpret, so more emphasis will be placed on this. The initial scheme of the short-range final stage of FPDS activity is shown below in Figure 2, but the rest of the endeavors will be connected with computer modeling and simulations of the virtual main planetary defense problem.

### 2.2. The Configuration of an Autonomous Space Vehicle for Recognition, Tracking, and Destruction of Asteroids

The first phase of the final includes reconnaissance, research, and analysis, and computer simulations of the attack. The second is the optimal attack. The third phase of the final consists of assessing the effects and possible repetition of the attack.

For this reason, the spacecraft has three stages; each spacecraft stage is autonomous and performs one attack phase.

The details of the first phase concern the remote recognition of the asteroid’s shape and composition using image spectrometry and autonomous sensors on robots for precise shape scanning and surface morphology. Currently, the problems of the visual classification of terrain for robot navigation based on AI methods are being considered [23]. Further sub-phases will determine the mean density and mass, sampling to determine the local porosity, and possibly the strength of the mixtures of components of the asteroid material. The collected samples will be used to study the dynamic properties of materials. On this basis, a dynamic, virtual mathematical–physical model of an asteroid will be prepared as a correctly posed initial-boundary value problem. The calculated effects, as the response to dynamic loading, allow the optimization of the attack. It will happen after launching a computer program related to the model and carrying out multi-variant simulations of the operation of available warheads and, for example, laser systems. The attack will start after the astronaut’s command and control loop decision. Due to the cosmic distances at this mission stage, a precise, autonomous navigation subsystem must be operational. An example is the successful completion of the mission to Comet Tempel 1 [24].

Details of the attack carried out in the second phase of asteroid control will be collected online from the autonomous network-centric sensors installed in the first phase on the surface and below the asteroid’s surface and in orbits around it. During recording, data will be used to evaluate the processes taking place in real-time. The network-centric sensor system will include numerous radio, radar, and optoelectronic sensors operating in various parts of the electromagnetic wave spectrum such as VIS, IR, and UV. Local sensors for in situ measurements of displacements, velocities, and accelerations, pressure and temperature sensors, and special seismic sensors will assess the effects of the online attack. It will be possible, almost in real-time, by comparing the measured values with the simulation results. Dust generated by the attack will not be an obstacle.

### 2.3. Motion and General Theory of Relativity—Quantum Metrology

Mathematical and physical dynamic modeling of the Future Planetary Defense System (FPDS) is a challenge. Modeling begins with selecting an appropriate physical theory that must consider the dynamics of network-centric autonomous spacecraft. In this case, the Resolutions of the XXX IAU General Assembly of 2018 formally force the modeling of motion in relativistic coordinate systems. Simultaneously, they caused the use of relativistic models of body dynamics, resulting from Einstein’s general theory of relativity, in the so-called post-Newtonian approximation [4,5]. Furthermore, large numbers of autonomous sensor systems and optoelectronic effectors are installed on-board spacecraft. Their operation should be modeled considering the relativistic models of electromagnetic wave and photon flux propagation resulting from Einstein’s general and special theories of relativity. This is imperative when their trajectories are close to the Sun or Jupiter. This kind of modeling is a new task compared to current modeling practice.

Additionally, in the case of autonomous space navigation, modeling requires the latest developments in space technologies at the level of quantum metrology. Many observations and research results from the area of interest, for obvious reasons, are still subject to many uncertainties. The next step is to select relativistic coordinate systems, barycentric for the solar system in its entirety and geocentric for Earth. They allow for setting the correct initial and boundary value problems (IVP and BVP). IVP and BVP are necessary for the proper modeling and simulation of the sequence of events that interests us. Interpretation difficulties require a more detailed discussion on modeling the relativistic problems in the solar system.

The principle of equivalence, which is also described in the general and special theories of relativity, applies to all physical laws. The principle states that the local properties of curved space in relativity should be indistinguishable from the properties of flat space–time in special relativity [4]. This is the path to the mathematical–physical algorithms of relativistic motion models. The concept of motion does not appear in relativity. However, by investigating the exact solutions of Einstein’s nonlinear equations, Einstein, with Infeld and Hoffmann (EIH) [4], proposed a methodology for a linear approximation of Einstein’s equations for the relativistic problem of two-body interaction in a weak gravitational field and at a low speed (concerning the speed of light). Infeld and Plebański summarized Einstein’s approach in the book *Motion and Relativity* [5] to solve the relativistic motion of N-bodies.

## 3. Initial Assumptions of the Future Planetary Defense System: Geometry, Reference Systems, Sensors, and Architecture Framework in Space Environment

### 3.1. Geometry and Reference Systems

The Future Planetary Defense System (FPDS) is intended to be an active multi-spherical layered system. The first layer of defense, the most distant, is a sphere with a radius r = 1.25 AU (covering a full solid angle of 4π steradians), with the center near the Sun in the relativistic Barycentric Celestial Reference System (BCRS). This part of the FPDS works against hazardous celestial objects using four rings of spacecraft covering the sphere. Dangerous objects can be small solar system bodies (SSSBs), which are from the main asteroid belt or deep space of the solar system or interstellar bodies (1I,2I) [13]. BRCS was generated as part of the application of general relativity using the method of linear post-Newtonian approximation, with the center in the barycenter of the solar system. Four rings of FPDS spacecraft are located on the mentioned sphere’s surface, each lying on a separate plane. The influence of the Kozai mechanism on asteroids and spacecraft has been ignored. Each subsequent plane is tilted from the ecliptic plane by an additional 45 degrees of ecliptic latitude.

The authors assumed that the second layer of defense, the medium range, will be a sphere with a radius of r = 900,000 km. This is close to the boundary but inside Earth’s gravity influence sphere (SOI), with the center in the relativistic geocentric celestial frame of reference (GCRS). Regarding the first spherical layer, it will consist of four space vehicle rings and will mainly work against near-Earth objects (NEOs), namely near-Earth asteroids (NEAs) and near-Earth comets (NECs). The most dangerous part of NEAs is a class of potentially hazardous asteroids (PHAs). The third spherical layer of protection, at very short distances, will be a system stationed on the Earth’s surface. This system will use the already operational and planned subsystems for the near-Earth Space Situational Awareness (SSA). This system, based on a network of large aircraft–rocket airports and systems for launching rockets from air-launch to orbit such as Stratolaunch System aircraft, allows for launching rockets with take-off weights of up to 230,000 kg. The FPDS system supported by a network of bases with human crews on Earth, the Moon, and Mars will have the astronauts in the control loop. In the area of interest of FPDS, a ring of space probes should appear in the ecliptic plane between Mercury and Venus, warning of events potentially dangerous for the Earth such as coronal mass ejections (CMEs) from the Sun’s surface. Planetary defense systems must be modeled in the solar system’s space environment with specific space weather including solar wind, solar flares, CMEs, and various radiation types. These space weather events can be a source of serious disturbance to synchronization processes. Among the complicated numerical modeling techniques, computational magnetohydrodynamics (MHD) can be found.

### 3.2. Proper Time, Atomic and Pulsar Clocks, SI of 2018, Autonomic Spacecrafts, Sensors, and Synchronization

The issue of sensors in communications subsystems such as radio frequency (RF), NASA Jet Propulsion Laboratory (JPL) Deep Space Network, or laser communications in space is critically dependent on the synchronization of their elements. The latter requires the operation of precise distance and angular tracking and positioning of the sensors onto the next spacecraft’s laser transceiver to establish and maintain communication and high data rate transmission. Since the position and speed must be very accurately known, an autonomous deep space navigation system requires an onboard atomic clock such as NASA’s JPL Deep Space atomic clock. With high stability at the 10^−15^ level, this atomic clock must be synchronized and coordinated with a pulsar clock, with stability at the 10^−18^ level [2,25]. All of the sensors above-mentioned including their relativistic corrections must be at the level of precision of the Quantum Metrology Standards (SI of 2018). During the modeling of the planetary defense chain’s components, some problems related to relativistic phenomena appeared on the solar system scale including synchronization processes. This work is an attempt to describe them.

### 3.3. Architecture Framework of FPDS and the DODAF—Department of Defense Architecture Framework

The term “planetary defense” encourages the use of the achievements of the military theory of Network-Centric Operations and Warfare (NCW). Within this framework, it is possible to correctly introduce the problems of collecting, transforming, and integrating data from a multi-level, hierarchical network of sensors located on spacecraft and distributed in space. In the next step, it is possible to process data into useful information that will be used to build knowledge of space situational awareness (SSA). Ultimately, people on Earth and astronauts in space, present in the feedback loop, will be able to run command chains to defend Earth against the impacts of small solar system bodies (SSSBs) or interstellar bodies (ISs) with the use of adequate actuators. A top–down methodology incorporating modified Marian Mazur models of the “Cybernetic Theory of Autonomous Systems” [10] was applied. This theory includes the analysis of cognitive problems in conjunction with decision-making problems. Because of a proposition to localize an autonomous spacecraft system as a top system of sensors in the deep space of the solar system, it is necessary to use relativistic space metrology tools when the SI of 2018 is already in force. This is necessary because of the universal relativistic coordinate system of the solar barycenter. The IAU resolutions require (starting from 1 January 2019) the use of general relativity theory to describe the space of the solar system and even local space around Earth in the weak field limit of the post-Newtonian equations of “slow” motion (PN). What does this mean for the clock synchronization subsystems, the entire planetary defense system, and its sensors? It implies very accurate and stable time signals from atomic and pulsar clocks and synchronization in the Poincare–Einstein convention. Finally, it provides the coordination of the joint operation of FPDS within the framework of special and general relativity.

The main risk to our civilization comes from small solar system bodies (SSSBs), with the specification of dangerous SSSB diameter, after Mark Boslough’s analysis [7], at 100 m and below. Due to a large number of small solar system bodies (SSSB) with diameters less than and equal to 100 m, the main problem of Earth’s planetary defense becomes complicated to solve. There are additional requirements for the FPDS architecture for clocks and sensors.

At the initial stage of building the computer concept of planetary defense (FPDS), the system architecture issue as the concept of the FPDS Architecture Framework modeling based on the ideas of the Department of Defense Architecture Framework (DoDAF v.2.2 and MODAF v.1.2 and NAFv.4) [12] was introduced. The process of even limited modeling of FPDS activities to support planetary defense activities and change management during routine service, and crisis begins after the definition of the FPDS-Architecture Framework (FPDS-AF). When designing architectural, geometrical, physical, and information mechanisms of the network-centric planetary defense system in space missions, the problem relies on our lowest level’s fundamental definition in the hierarchy (i.e., the autonomous subsystem model, according to Mazur’s description) [10]. It is necessary to emphasize (a) the role of sensors in the information channel for data collection and exchange, telemetry, and system synchronization, and (b) the role of actuators in the energy channel and the collection, generation, and accumulation of energy, and finally (c) the role of the local center of command and control with supercomputer support, and with the appropriate relational database of the big data class for mission co-management and the creation of space situational awareness (SSA). The FPDS-Architecture Framework is an infrastructure resource of organizational, financial, material, and system tools to prepare and build new concepts of planetary defense capability. It is also used for assessment, from an operational, technical, and systemic point of view, in a mode analogous to Network-Centric Operations and Warfare (NCW). It generates an analysis of the state of space situational awareness (SSA) and future activities that create or restore infrastructure resources. Briefly summarizing, the data flow from FPDS-Architecture Framework allows for the use of this updated knowledge for long-term management and the control of future operations during planetary defense missions.

The first task in our concept’s computer modeling is to prepare mathematical–physical and numerical models, both classic and relativistic. After that, we must simulate key physical processes accompanying emerging threats and combat these threats using massive resources of directed energy, remembering that the effectiveness of planetary defense depends on changing the SSSB’s momentum, for example, during collision momentum exchange.

The second task is to resolve the problems of integration of subsystems and the connections between them. The next steps include identifying and modeling the new system capabilities, then formulating software requirements, analyzing the costs, and preparing the model of financial service for the mission by estimating the number of subsystems needed to implement the mission.

The third task is to identify, analyze, model, and simulate an autonomous subsystem of sensor interconnected networks using quantum space metrology parameters with relativistic corrections.

The high energy-demanding physical planetary defense processes coincide with information processes such as decision-making processes, which have minimal energy consumption. All processes referred to here are “encapsulated” in various categories of extensively distributed sensor networks. These networks must work and be synchronized (a common problem of special relativity theory) considering relativistic corrections for the extremely precise (almost exact) quantum space metrology tools. How to carry this out is a central problem raised in this paper.

### 3.4. Future Planetary Defense System Closer to the Main Asteroid Belt

A FPDS closer to the main asteroid belt means that the perimeter of the system is equal to 1.25 AU. Early analysis of threats to Earth from incoming asteroids and comets treated them as single, unique events. The focus was on single space missions, which included the issues of the launch phase, precise navigation, and the arrival of a space vehicle equipped with the appropriate set of actuators (e.g., classic warheads, nuclear warheads, or a super-laser with extraordinary energy and power) directed at the target, which is sufficient for active interactions with “small” celestial bodies directly threatening the Earth.

An infrared telescope, the Wide-field Infrared Survey Explorer (WISE), has been operating in space for the past 10 years (2009–2019). WISE has discovered 19 new comets and over 33,500 new asteroids including 120 NEOs relatively close to Earth. In the final phase of the WISE mission, NASA began a new mission called Near-Earth Objects WISE (NEOWISE) [14,15], which has observed 153,726 solar system objects. Before hibernation, this spacecraft delivered characterizations of 158,000 minor planets including more than 35,000 newly discovered objects. Starting in December 2013, the post-hibernation NEOWISE mission was anticipated to discover 150 previously unknown near-Earth objects and to learn more about the characteristics of 2000 known asteroids. NEOWISE’s automated detection software, WISE Moving Object Pipeline Subsystem (WMOPS), detected only a few objects smaller than 100 m in diameter because doing so requires five or more detections. The average albedo of asteroids larger than 100 m discovered by NEOWISE was 0.14 [14,15]. Research from WISE/NEOWISE missions involving the search for small solar system bodies using IR sensors revealed the correct scale of small body threats (see Figure 3) [15]. This prompted us to propose research and implement a new future planetary defense system (FPDS) closer to the main asteroid belt.

Step by step, scientific and research missions that have successfully navigated into the solar system’s more in-depth and deeper areas were beneficial for this approach. The initial question then arose concerning the assessment of the “real effects” of such interactions: is there even a slight possibility of preventing a small celestial body’s catastrophic collision with Earth?

In the planetary defense case, the team started in the early 1990s [16,17,18,19], preparing the hydrocodes of high energy density physics for cosmic bodies for high-velocity collision modeling [17,18]. For several decades, many cases of catastrophic space impacts and collisions have been modeled in this field, with the hydrocodes HEFPM-G with gravity. Many visualizations and computer movies with additional tools have helped analyze and understand the processes and effects of collisions [19]. These results were sufficiently positive to start planetary defense system modeling [20,21]. This means that there were opportunities to change the path or even destroy specific categories of asteroids and comets. Early answers were somewhat optimistic, and a change in the linear velocity component with DeltaV = 1cm/s of the known asteroid 4179 Toutatis should be enough [26]. However, that was only the beginning of the necessary actions. For this reason, we are now able to show the qualitative and quantitative effects of dynamic impacts on a selected class of SSSBs, and even out-of-solar system asteroids (e.g., 1I/’Oumuamua) or comets (e.g., 2I/Borisov) [13,27].

Our strikes (i.e., planned collisions) are contact strikes, so the precise navigation of the spacecraft is critical. It requires the correct selection and construction of several subsystems in which the necessary sequences of activities to accomplish the mission are carried out.

### 3.5. Precision of Navigation

There are many problems with precision. The first class is associated with a space metrology measuring system for determining the position and velocity of a spacecraft in the general relativity frame, where the quantum systems of laser radars must work with space metrological precision.

The second class is associated with the synchronization processes and data transmission, which is possible only using special relativity theory. Formally, the synchronization process can occur in space, in which the special theory of relativity is in force, along straight lines and along with closed tracks back and forth. Synchronization cannot take place along beams of photons running near large masses, for example, to the Sun or Jupiter, which is a problem related to bending light beams in gravitational fields.

It seems, however, that data transmission can take place in beams of light photons passing “close” to large masses. Then, the essence of data transmission is the flight of a photon bundle, where we can reconstruct the states of the photons located in the bundle with each other without worrying about the time of flight.

The third is associated with the accurate, relativistic model of the solar system’s motion, the fundamental problem of general relativity theory.

The fourth is related to the construction of a precise relativistic model of spacecraft motion.

Suppose navigation with relativistic precision leads close to the target, and after that, to the collision mission. In this case, the decision chain and “fire chain” sequence begins, forming a planetary defense chain, leading to the selected form of interaction with the small solar system body. An example of a collision mission was “Deep Impact.” Interestingly, Deep Impact used the autonomous optical navigation (AutoNav) software system [24] to guide the Impactor spacecraft to intercept the nucleus of Tempel 1 at a well-illuminated location, visible from the fly-by. The fly-by spacecraft used similar software to determine its comet-relative trajectory and to provide the attitude determination and control system (ADCS) with the relative position information necessary to point the high-resolution imager (HRI) and medium resolution imager (MRI) instruments at the impact site during the encounter [14].

### 3.6. Top–Down Methodology of C4ISTAR to Obtain Network-Centric Capabilities with Optimal Geometry, Functionality, and Autonomy with Decision-Making Model

Years ago, Norbert Wiener emphasized that we must first have “know-what” before we tackle “know-how” (i.e., the technological problem). It suggests giving preference to a top–down methodology in the first research phase of the Future Planetary Defense System (FPDS) computer design. The essence of this proposal, which is based on the NCW pattern, is the transformation process. From the space missions of individual spacecraft (individual platforms) with various types of sensors and effectors on board, which perform functions related to planetary defense with a platform-centric approach, transformation to the system of many spacecraft connected in a network and integrated into network-centric systems with new cooperation algorithms, is crucial. As a result, it needs to obtain several network-centric capabilities such as the following:(1)New capabilities of the broadband space digital data transmission network with the appropriate quality of service including reliability and security, guaranteeing the full interoperability of spacecraft and their subsystems: A Space INTERNET with a modern mobile protocol;(2)Network capabilities of space sensor carriers of many intelligent classes including autonomous sensors in space, forming the combined Space MEGA SENSOR of the future planetary defense system;(3)Computational network capabilities of the spacecraft, which are nodes of the defense system, with supercomputing centers on board with mass-parallel computing architecture and performance of many tens of Teraflops for local data processing, information, and knowledge gathering for space situational awareness (SSA) databases, forming a joint Space TERA GRID; and(4)Network capabilities of new kinetic and electromagnetic actuators for the precise destruction of small celestial bodies that pose a threat to Earth. Our space warheads will be able to change the trajectories of small celestial bodies on collision courses with Earth, and they will serve as a network through connection with the Space MEGA SPACE-WARHEAD. A network of other systems is also possible (e.g., directed energy weapons based on Space MULTIDEW combined laser systems).

This list of potential capabilities suggests that the Future Planetary Defense System (FPDS) must be a distributed network-centric system (see Figure 4). In this scheme, our nodes use the idea of Baran’s work [28], which he named distributed communications.

A crucial part of FPDS distributed network sensors—at the highest level of integration, being just a set of sensors on all spacecraft in the solar system’s inner space—is that it has high power nodes with supercomputers on board and with sophisticated computational applications. This primary FPDS subsystem will also create an extensive computing system with multi-TERAFLOPS of power in space. It will be a subsystem for processing useful informational data from sensors with the potential to model events in less time than those that occur in real-time; it will also serve to expand the space situational awareness (SSA) of upcoming problems and support the operators’ knowledge during the commissioning and implementation of the planetary defense decision chain (see Figure 5).

The concept of autonomous sensor systems in spacecraft means that the classic definition of sensors as a transducer subsystem is insufficient. The top–down methodology of the FPDS concept requires the following modified Marian Mazur definition of an autonomous system with on-board sensors and actuators [10]: an autonomous system is a system with the capability to control itself and the ability to counteract the loss of its control capabilities.

Figure 6 shows the information and energy channels in autonomous spacecraft.

For the subsystem node of the entire FPDS (i.e., an autonomous spacecraft with sensors and effectors), the definition can be supplemented with the following requirement that it should be a spacecraft capable of remaining in the environment for as long as possible.

This means that the autonomous system must contain the relevant components as subsystems, and above all, the following:(1)intelligent sensors (i.e., components for collecting information from the environment);(2)special sensors for system synchronization supported by atomic and pulsar clocks;(3)C4ISTAR as local management and command center, authorized when needed by people, and collecting, processing, storing, and transferring information about space situational awareness (SSA); it must contain a homeostat and homeostasis stabilization subsystems, and duplicate supercomputers must support everything above-mentioned; and(4)Effectors (i.e., components to affect the environment).

Effectors should receive data locally from the sensors as above-mentioned. Globally, they should receive data, information, and commands authorized, if necessary, by people, from C4ISTAR, for which an information channel is needed. C4ISTAR decides and determines which of the possible interactions are to occur and allocates energy resources, enabling the work necessary for the effectors to perform the interaction; for this, an energy channel is needed that includes the following:(5)technical systems for obtaining energy from the environment;(6)local energy sources (energy and power generators);(7)energy storage and conversion batteries; and(8)technical energy distribution and processing systems.

Since we are preparing a future system, we are interested in two categories of problems: cognitive, related to data collection (by flocks of intelligent sensors) and decision-making. For the requirements of the Future Planetary Defense System, we can directly use the following slightly modified NCW tenets associated with data collection and information processing [6,12]:

Tenet 1: A robustly networked spacecraft improves information sharing.

Tenet 2: Information sharing and collaboration enhance the quality of information and shared space situational awareness (SSA).

Tenet 3: Shared space situational awareness (SSA) enables self-synchronization.

Tenet 4: Self-synchronization, in turn, dramatically increases mission effectiveness.

This multi-stage process is shown in Figure 7.

As a decision-making model at the early stage of building our preliminary computer concept of the Future Planetary Defense System, we chose Boyd’s Observe–Orient–Decide–Act (OODA) loop. NCW and Command and Control (C2) systems architecture theory adopted OODA without any change.

The OODA loop is transparent and straightforward, so it will be useful at a new computer concept stage. For military practitioners, the transition from platform-oriented systems to integrated and “simple” network-oriented systems is not sufficient. We need to move from a network-oriented approach to a knowledge-based approach to defense systems [6]. Figure 8 shows the new chain of the knowledge-based approach. It will be much easier to close the OODA loop at the top of the chain.

In fact, in the FPDS concept, we will have at least three levels of hierarchy. The lower level is the hierarchy of autonomous sensors and actuators on spacecraft. The middle level is the hierarchy of autonomous spacecraft and the higher level is the hierarchy of FPDS spheres.

FPDS spacecraft are dynamic systems with complex characteristics, especially during pursuit maneuvers with the use of rocket engine thrust and change in direction, nonlinear vibrations appear. For this reason, during the analysis of the issues of simulation and control of autonomous operations of space vehicles, problems arose with their proper modeling and the development of signals for space vehicle actuators while solving systems of nonlinear dynamics equations of bifurcation problems [29,30] and used neural network algorithms [31].

As the leading technologies for connecting nodes in the system, we propose laser and millimeter-wave (MMW) technologies, but we will use radio frequency (RF) technologies. By IAU definition, small solar system bodies (SSSBs) are solar system objects that are neither planets, dwarf planets, nor natural satellites. Asteroids with diameters of 100 m, or slightly smaller, also being SSSBs, pose a high risk to the safety of the increasingly densely populated Earth. Such bodies are already hazardous when they hit the surface of the Earth [7,21].

The arsenal to combat threats from such celestial bodies, using energy in the range of 50–100 MT TNT, is already available using existing technology. It is possible to use this kind of actuator through contact or short-range kinetic interactions. We can apply classic explosives or nuclear warheads, and soon we may be able to use extremely high-powered lasers. However, nuclear explosions can be a source of further threats to space vehicles, synchronization processes of space vehicles, and their electromagnetic compatibility (EMC). With repeated action, electromagnetic pulses of nuclear explosions in space may appear from interactions with the plasma of the ionized matter of the bomb structure and with the plasma of the ionized remains of the asteroid. Another threat is a system-generated electromagnetic pulse (SGEMP) or spacecraft charging (SCC). However, it is not the end of the space mission; synchronized sensors are crucial in the next stage of the mission. After the attack, we will need a select category of sophisticated sensors to evaluate the attack results and confirm the success of the mission. To evaluate certain parameters, we will prepare a simple scenario and simulation of the deflection mission. For this reason, we must prepare a precise computer experiment to determine how our weapons will interact with the asteroid. We need to assess what deformations of the attacked asteroid we can expect and which sensors to choose to measure, from a safe distance, the level of damage.

## 4. A Concept of Future Planetary Defense System Synchronization with an Autonomous Sensor Hierarchy

The synchronization of the Future Planetary Defense System (FPDS) will be difficult due to its environment and the extreme demands placed on the technologies used. The radius of action at astronomical distances found in the solar system and the complex tasks of combating small celestial bodies make the degree of difficulty higher than we have dealt with so far. Modeling synchronization processes is also difficult because it requires learning approximate methods of describing quantum phenomena and approximate methods of describing relativistic phenomena. Autonomous quantum sensors such as atomic clocks and interferometers have just arrived onboard spacecraft in deep outer space and come from the microworld. Modeling and simulations of their internal operation are carried out using mechanics and quantum electrodynamics because they operate on a subatomic scale. Meanwhile, modeling the cooperation of these instruments within the planetary defense system’s framework covers the largest known ranges on the cosmic scale. Everything will have to be rebuilt from scratch in this area, both the instruments and their models. Future technologies will have to be planned and pre-tested in computer models to new requirements. In turn, the presence of optical communication technology using photons, where quantum systems with laser transmitters and receivers of coherent detection (with homodyne and heterodyne) or direct detection (photon counting) are combined with the use of semi-classic models of shot-noise. Secure communications in space with the distribution of quantum cryptographic keys (QKD) is a future technology with applications in space. Remote sensing with classic measurements using electromagnetic waves and quantum (photon) instruments in the field of quantum metrology (e.g., laser radar systems) will be used in space for detection, parameter estimation, and imaging of small celestial bodies [18]. The entire class of autonomous sensors and actuators aboard the class of pico- and nanosatellites will be used for in situ measurements on the surfaces of small cosmic bodies. We are talking about a set hierarchy of synchronization problems from the highest to the lowest level.

The highest level of the hierarchy concerns the synchronization of the entire FPDS (i.e., the coordination of events necessary for the proper interaction of all three system layers). The system’s first layer is a sphere with a radius r = 1.25 AU around the Sun. The second layer of the system is a sphere with a radius r = 900,000 km around the Earth. We place the third layer of the system on Earth’s surface, covering the space around Earth up to the height of the geostationary orbit. The essence of the problem is the synchronization of activities aimed at linking events into a cause-and-effect chain subject to the rules of decision making and issuing commands with people’s participation in feedback loops. Synchronization processes must accompany the highest synchronization and event coordination at an intermediate level in this hierarchy. Synchronization signals must be exchanged between autonomous spacecraft, just like process synchronization signals between autonomous quantum sensors aboard these spacecraft. This basic level of the autonomous sensor hierarchy requires prior synchronization of atomic clocks at an intermediate level of synchronization (i.e., onboard spacecraft). Pulsar clocks onboard the spacecraft (i.e., at an intermediate level of the hierarchy) will additionally support all synchronization processes due to their long-term stability. Ultimately, after overcoming these difficulties, the clocks of the individual autonomous sensors will be synchronized. However, before that, these processes must be modeled and simulated in computers.

Figure 9 is a snapshot of the interactive 3D visualization of Asteroid Space Asterank [32].

The modeling techniques used for these purposes are from various fields of science (e.g., computational physics and astrophysics in relativistic frames of special relativity and general relativity and computational quantum mechanics). We will model the clocks’ synchronization in the sense of the Poincare–Einstein convention and FPDS synchronization events as a coordination of the joint operation of the entire FPDS with the self-synchronization effect of space situational awareness (SSA) knowledge with people in the loop.

In Section 5, the simulation results are given of the final part of asteroid deflection, a simple scenario against 100 m diameter asteroids. However, we first explain the relativistic model of motion.

## 5. General Relativity Theory and Post-Newtonian Approximation (PN)—Motion in PN Reference Systems within the Framework of General Relativity

The most fundamental theory of classical physics is general relativity. Following the international recommendations of the IAU (International Astronomical Union), we will describe events occurring in space in the solar system as well as near Earth. Therefore, at the stage of the initial computerized concept of FPDS construction, as part of the preparation of a mathematical–physical and numerical model for the computer simulation of events occurring during the operation of the Future Planetary Defense System, the theoretical description must be created within Einstein’s general relativity.

The Einstein field equations of the astrophysical applications inside the solar system are:(1)$$G_{\mu \nu }=\frac{8\pi }{c^{4}}GT_{\mu \nu }$$
where: $G_{\mu \nu }$—is Einstein’s tensor; $g_{\mu \nu }$—is the metric tensor (−, +, +, +); *c*—is the speed of light in a vacuum; G—is the gravitational constant; and $T_{\mu \nu }$—is the stress–energy tensor.

In general relativity theory, we must solve the equations of motion and field equations simultaneously, which means that the problem of motion cannot be strictly solved [1]. Only the approximate method can be used, which consists of the power series expansion of all functions appearing in the field equations against the negative powers of the speed of light. This problem was first solved in the work of Einstein, Infeld, and Hoffmann in 1938 [2]. In 1960, Infeld and Plebański finished solving the problem of “Motion and Relativity”, in the spirit of Einstein, giving us, derived by the EIH approximation method, the relativistic n-body equations of motion. T.D. Moyer [33] used the Lagrangian function from the work of Infeld and Plebański to obtain the approximate relativistic equation of motion for determining the ephemeris in the solar system. This is the procedure that led to the Newtonian equation of motion with post-Newtonian corrections. Through numerical integration, it is possible to prepare the ephemeris for SSSBs and spacecraft of the FPDS within the solar system. The equation from Montenbruck and Gill’s [34] work is the same as that from the work of Moyer [33]:(2)$$\vec{\ddot{r}}=-\frac{GM}{r^{2}}\left(\left(4\frac{GM}{c^{2}r}-\frac{v^{2}}{c^{2}}\right)\cdot {\vec{e}}_{r}+4\frac{v^{2}}{c^{2}}\left({\vec{e}}_{r}\cdot {\vec{e}}_{v}\right){\vec{e}}_{v}\right)$$

Moyer, in the 1960s, prepared his work [33] based on Infeld’s work from Infeld and Plebański’s monography [5]. In Spier work [35]:$$\vec{\ddot{r}}=-\mu_{c}\frac{\vec{r}}{r^{3}}+\underset{i}{\sum }\mu_{i}\left[\frac{{\vec{r}}_{ic}}{r_{ic}{}^{3}}-\frac{{\vec{r}}_{ip}}{r_{ip}{}^{3}}\right]+\vec{r}\left(OBL\right)+\vec{r}\left(SRP,AC\right)+\vec{r}\left(MB\right)+\vec{r}\left(IOBL\right)+\vec{r}\left(GR\right)$$
where

$\vec{\ddot{r}}$ is the acceleration of the spacecraft;

$\mu_{c}$ is the gravitational constant of the center of integration, ${\mathrm{km}}^{3}/s$;

$\mu_{i}$ is the gravitational constant of the body i, ${\mathrm{km}}^{3}/s^{2}$;

$\vec{r}$ is the position of the spacecraft relative to the center of integration in 1950.0 Earth equatorial rectangular coordinates;

${\vec{r}}_{ic}$ is the position of the body i relative to center of integration in 1950.0 rectangular coordinates; and

${\vec{r}}_{ip}$ is the position of the spacecraft relative to body i in 1950.0 rectangular coordinates. $$\begin{array}{cc} {\vec{\ddot{r}}}_{i}\left(j\right)=\frac{\mu_{j}\left({\vec{r}}_{j}-{\vec{r}}_{i}\right)}{r_{ij}{}^{3}} & \left\{-\frac{4}{c^{2}}\phi_{i}-\frac{1}{c^{2}}\phi_{j}+{\left(\frac{\dot{s_{i}}}{c}\right)}^{2}+2{\left(\frac{\dot{s_{j}}}{c}\right)}^{2}-\frac{4}{c^{2}}\dot{{\vec{r}}_{i}}\cdot \dot{{\vec{r}}_{j}}-\frac{3}{2c^{2}}{\left[\frac{\left({\vec{r}}_{i}-{\vec{r}}_{j}\right)\cdot {\dot{\vec{r}}}_{j}}{r_{ij}}\right]}^{2}\;+\frac{1}{2c^{2}}\left({\vec{r}}_{j}-{\vec{r}}_{i}\right)\cdot {\vec{\ddot{r}}}_{j}\right\}\\ & +\frac{1}{c^{2}}\frac{\mu_{j}}{r_{ij}{}^{3}}\left[\left({\vec{r}}_{i}-{\vec{r}}_{j}\right)\cdot \left(4\dot{{\vec{r}}_{i}}-3\dot{{\vec{r}}_{j}}\right)\right]\dot{({\vec{r}}_{i}}-\dot{{\vec{r}}_{j}})+\frac{7}{2c^{2}}\frac{\mu_{j}{\vec{\ddot{r}}}_{j}}{r_{ij}} \end{array}$$


$r_{ij}$ is the coordinate distance between bodies i and j;

$(\dot{s_{i}{)}^{2},}(\dot{s_{j}{)}^{2}}$ is the square of the velocity of bodies i and j, respectively;

$\phi_{i}$ is the Newtonian potential at body i;$\phi_{j}$ is the Newtonian potential at body j.

And:$${\vec{\ddot{r}}}_{j}=\underset{m\neq j}{\sum }\frac{\mu_{m}\left({\vec{r}}_{m}-{\vec{r}}_{j}\right)}{r_{mj}{}^{3}}$$

For example, for a circular orbit $\frac{GM}{r}=v^{2}$, and a relativistic correction of acceleration, a simple equation is given:(3)$$\vec{\ddot{r}}=-\frac{GM}{r^{2}}{\vec{e}}_{r}\left(3\frac{v^{2}}{c^{2}}\right)$$

When the XXXth General Assembly of the. IAU (2018) recommended a reference system in the framework of general relativity in the post-Newtonian approximation, it became an imperative for our concept because any mathematical and physical computer modeling of the system is related to the selection of an appropriate reference system. Fortunately, for interactions of very small contact bodies, we can use classical mechanics of continuous media.

## 6. The Threat from a 100 m Diameter Asteroid Impact on Earth Estimation

Thanks to the analysis conducted by Mark Boslough [7], the main risk to Earth’s population from small solar system bodies (SSSBs) was estimated for a 100 m diameter body. To estimate a realistic threat to Earth’s population from a 100 m diameter asteroid impact with a mathematical and physical model, we used the ImpactEarth! Portal available in the network [20] based on an article by G.S. Collins, H.J. Melosh, and R.A. Marcus, Earth Impact Effects Program: A Web-based computer program for calculating the regional environmental consequences of a meteoroid impact on Earth [21], giving approximate results of modeling, with a high level of confidence. Initial data, characteristics of the asteroid (Table 1), and the calculation results are provided (Table 2).

The simulation results indicate that an area exceeding 300 km^2^ would be seriously damaged.

## 7. The Final Part of the FPDS Deflection Mission Modeling and Simulations

To verify the possibility of destruction, damage, or redirection to another orbit of 100 m in diameter asteroids using technology available on Earth, we decided to use our hydrocode with the K. Jach “free particles” method [21] to simulate physically faithful interactions on stone asteroids with a density of 2400 kg/m^3^ and a diameter of 100 m. We used three variants of events with targets within hundred meters in the order: (1) asteroid onto asteroid impact; (2) conventional explosive charge detonation deflection; and (3) hypothetical laser deflection in some simplified configurations.

We prepared the full mathematical–physical model [18] starting with a hydrostatic model of an asteroid 100 m in diameter. The 2.5D cylindrical, axially symmetric (r,z) coordinate system with a spherical asteroid was applied, as seen in Figure 10.

We did not include the strength of the material. The spherical symmetry of the undisturbed asteroid was assumed.

The initial density distribution $\rho_{0}\left(R\right)$ inside the asteroid is as follows:(4)$$\rho =\rho_{0}\left(R\right),\;0\;<\;R<R_{0}\;$$
where *R*_0_ denotes the asteroid radius at the initial moment of time.

The initial distribution of Grüneisen parameter γ inside the asteroid is as follows:(5)$$\gamma =\gamma_{0}\left(R\right),\;0\;<\;R<R_{0}\;$$

The Grüneisen parameter *γ* is assumed to be undisturbed.

The equation of the state of the asteroid’s material is consistent with the following form of the Mie–Grüneisen equation of state, in which the first part is similar to the pressure–density relationship of Murnaghan, and the second part is the Grüneisen term:(6)$$p-p_{0}=\frac{K_{0}}{K_{0}^{\prime }}\left[{\left(\frac{\rho }{\rho_{0}}\right)}^{K_{0}^{\prime }}-1\right]+\gamma \rho \left(E-E_{0}\right)\;$$
where p is the pressure; ρ is the density; and E denotes the specific energy per unit of mass. $K_{0}$ denotes the initial bulk modulus, which depends linearly on the undisturbed pressure $p_{0}$:(7)$$K_{0}\;\mathrm{is}\;\mathrm{identical}\;\mathrm{to}\;K\left(p_{0}\right)=K_{00}+K_{0}^{\prime }\cdot p_{0}\;$$

For silicate rock:(8)$$\begin{array}{c} K_{00}=1.5\times 10^{11}\;\mathrm{Pa}\\ K_{0}^{\prime }=3.3 \end{array}\}\;$$

The initial and undisturbed pressure p_0_ are determined by the hydrostatic equation:(9)$$dp_{0}=-\frac{Gm_{0}\left(R\right)}{R^{2}}\rho_{0}\left(R\right)dR\;$$
where $G=6.6743\times 10^{-11}\;m^{3}\cdot {\mathrm{kg}}^{-1}s^{-2}$ is the gravity constant and $m_{0}\left(R\right)$ is the partial mass within an undisturbed sphere of radius R.
(10)$$m_{0}=4\pi \int_{0}^{R}\rho_{0}\left(x\right)x^{2}dx\;$$

The density of the impactor is assumed to be equal to 2400 kg/m^3^.

The construction of a hydrodynamic mathematical–physical model of the asteroid deflection process from the collision course begins with the preparation of an appropriate system of partial differential equations that describe the relevant conservation laws. The numerical model is a hydrocode of the free particle method (HEFPM) [21].

The following set of equations of mass, momentum, and energy conservation was used:

Mass:(11)$$\frac{d\rho }{dt}+\rho \left(\frac{\partial u}{\partial r}+\frac{\partial v}{\partial z}+\frac{u}{r}\right)=0$$

Momentum:(12)$$\rho \frac{du}{dt}=-\frac{Gm_{0}\rho }{R_{}^{2}}\;\frac{r}{R}-\frac{\partial p}{\partial r}$$
(13)$$\rho \frac{dv}{dt}=-\frac{Gm_{0}\rho }{R_{}^{2}}\;\frac{z}{R}-\frac{\partial p}{\partial z}$$

Energy:(14)$$\rho \frac{dE}{dt}=-p\left(\frac{\partial u}{\partial r}+\frac{\partial v}{\partial z}+\frac{u}{r}\right)$$
where
(15)$$\frac{d}{dt}=\frac{\partial }{\partial t}+u\frac{\partial }{\partial r}+v\frac{\partial }{\partial z}\;$$

The boundary condition (pressure) on the free surface of the asteroid is equal to zero:(16)$$p\left(R_{As}\right)=0\;$$

The equations of this problem can only be solved by numerical means.

### 7.1. The Results of the Computer Simulation of 100 m Diameter Asteroid Deflection by Impact of Another Asteroid 13.5 m in Diameter with a Velocity of 10 km/s and 3 km/s

Figures (Figure 11, Figure 12, Figure 13 and Figure 14) show the visualization of the hydrocode simulation results, as the time evolution of asteroid response in asteroid onto asteroid impact. The colors are related to the pressure scale.

### 7.2. The Results of the Computer Simulation of an Asteroid Deflection Using a Conventional Explosive with an Octagen Charge Mass of 7100 kg

Figures (Figure 15, Figure 16 and Figure 17) show the visualization of the hydrocode simulation results as the conventional explosive. The colors are related to the pressure scale.

### 7.3. The Results of the Computer Simulation of the Deflection of a 100 m Diameter Asteroid Using Giant Laser Pulse

Figures (Figure 18, Figure 19 and Figure 20) show the visualization of the hydrocode simulation results as the hypothetical laser deflection. The colors are related to the pressure scale.

## 8. Isotropic Distribution of Sensors on the Ring’s Planes

We focused on the outer spacecraft circular ring with a radius of 1.25 AU. In the first approximation, on each plane of the ring, we obtained the isotropic spacecraft distribution, and by using the appropriate autonomous spacecraft, we obtained the proper number of degrees of arc (e.g., with an initial number of spacecraft of 24, 15 degrees of arc; for 36 spacecraft, 10 degrees of arc; and for 72 spacecraft, 5 degrees of arc).

In general relativity (GR), the gravitational field affects and determines the metric laws of space–time. For measurements, we will need clocks in inertial coordinate systems and “practically rigid bars” as in special relativity (SR). Each spacecraft will have a set of three to four atomic clocks for use in deep outer space.

Our concept of Future Planetary Defense System includes the detection and continuous tracking of threats from space of the solar system (from the main asteroid belt and especially from the near-Earth objects (NEOs)), together with a combat strategy based on the Network-Centric Operation and Warfare theory with the flock algorithms [6].

In the inner part of the solar system, at a distance of r = 1.25 AU, we have to create the four rings of small autonomous spacecraft equipped with warheads. The spacecraft can operate in the four planes with a common equinox vector. For the initial orbit calculations and visualizations, we used the Asterank Portal prepared by Ian Webster [32]. Four planes deviate from the fundamental ecliptic plane: the first plane by ecliptic latitude is b = 0° (Ring of Spacecraft b0), the second plane by ecliptic latitude is b = 45° (Ring of Spacecraft b45), the third plane by ecliptic latitude is b = 90° (Ring of Spacecraft b90), and the fourth plane by ecliptic latitude is b = −45° (Ring of Spacecraft b-45), as seen in Figure 21, Figure 22, Figure 23 and Figure 24 [32].

## 9. A Few Preliminary Comments on the Modeling of Sensor Synchronization in the FPDS

High-resolution and high accuracy measuring tools and methods came out of quantum metrology research.We focused on the outer circular ring with a radius of 1.25 AU with the initial number of spacecraft for modeling ranging from 24 to 72.Each spacecraft will contain a set of three to four atomic clocks for use in deep space.Each spacecraft will have a pulsar clock system.There will be a laser transceiver system on every spacecraft.Measurement, synchronization, and coordination will be based on the exchange of laser light pulses along the ring, on both sides, from the primary spacecraft.In general relativity (GR), the gravitational field acts and determines the metric laws of space–time. For measurements and synchronization, we will need clocks in inertial coordinate systems and “practically rigid bars” such as in special relativity (SR).Our task is to analyze, through computer simulations, the values of the gravitational field in the neighborhood of the FPDS space vehicle ring and the study of laser pulse propagation phenomena to meet STW conditions with high accuracy (the velocities of massive bodies in the solar system are very low compared to the speed of light).

## 10. Summary

The success of the Planetary Defense System depends on the proper autonomous sensor system at the space metrology precision level. This system of systems must work with the relative and special theories frameworks. The spherical navigation and precise positioning in the framework of general relativity require superior clocks because our sensors must find navigation positions by estimating the distance from three known fixed points measured by, for example, the flight time of a laser pulse. We need incredibly precise clocks for these purposes with atomic, radio pulsar, and X-ray pulsar time as well as excellent clocks for system synchronization, coordination and correlation; perfect reference time scales; and perfect clock evaluation algorithms. We illustrated the concept and presented some key recommendations for such a relativistic metrology subsystem.

New ideas and new proposals for computer models and programs must appear in this field of space techniques due to two groundbreaking changes made last year.

The first change is the obligation to implement the relevant resolutions of the 30th General Assembly of the IAU (International Astronomical Union) from 1 January 2019 regarding the use of relativistic reference systems.

The second groundbreaking change concerns the SI system, since it was introduced on 1 September 2019, the so-called “The New SI System of Units” or “SI of 2018”, which introduced quantum metrological standards.

New functional models of computer space mission simulators will now apply for autonomous spacecraft and their autonomous sensors. On a macro-scale, they are compatible with new mathematical and physical models based on general relativity. On a micro-scale, they will be compatible with models based on quantum mechanics and quantum electrodynamics.

The defense system proposal is entirely new and is based on the latest achievements and “know-how” of the technology of large military network-oriented (network-centric) systems. It is a multilayered model of the Future Planetary Defense System.

Meanwhile, asteroids and comets that, because of collisions in the Main Belt of asteroids, may be on collision courses with Earth, those that will be several at once, and those that do not come from the solar system such as Oumuamua or Borisov-2 may pose a multiple threat.

New programs for computer simulation of future large planetary defense systems must be based on models that accurately reproduce the physics of phenomena at the level of quantum metrology precision. We are trying to improve the descriptions in the text because we described new models modified by us to planetary defense. We obtained accurate results within the accepted models. According to these simulations, it will be possible to assess, for example, the possibility of using selected sensors to analyze the impact of planned impacts on asteroids or comets.

The level of accuracy of simulations using our hydrocodes checked, for example, for the time of shock wave propagation to Earth’s antipodal point, was better than one promille. Modern technologies applied in favorable conditions allow us to fight asteroids with diameters up to 100 m. Solutions to fighting larger celestial bodies by repeated use of classic attacks leading to the gradual destruction of asteroids and comets will be presented in subsequent articles. The problems of using nuclear charges for the same purposes require separate treatment.

## Figures and Tables

**Figure 1 materials-15-01752-f001:**
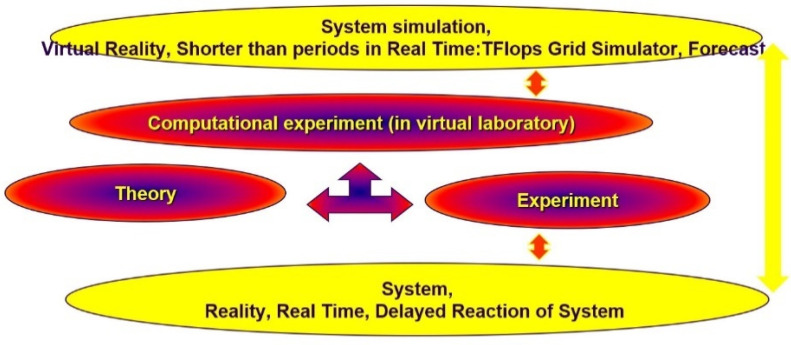
Paradigm of research in science (in red) with ideas of FPDS simulation (in yellow).

**Figure 2 materials-15-01752-f002:**
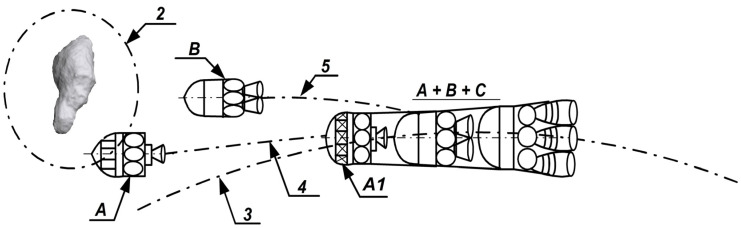
The short-range final part of FPDS activity with the use of single autonomous spacecraft. This mission is finished with three stages of autonomous spacecraft using specific loops of activity. For our purposes, it is possible to extend the OODA loop developed by Colonel John Boyd, which is the cycle: observe–orient–decide–act, by the cycle observe–orient–decide–act-check and repeat the loop OODA. The potentially hazardous object (PHO) here is a near-Earth asteroid (NEA), which is on a collision course with Earth. The asteroid was detected and tracked previously by the A1-MEGA SENSOR, part of the A sub-spacecraft. It is the third stage of autonomous spacecraft for the detection and observation of the close space of NEA orbiting during mission action. Pico- or femto-space probes for surface landing and local NEA research may also be used. NEA can be attacked by using the B-MEGA SPACE-WARHEAD actuator subsystem as the second stage of autonomous spacecraft for the destruction of the hazardous NEA. The C “mother”, as the first stage of autonomous spacecraft, observes the mission results, securing this action of the FPDS. Spacecraft Stage C, or “mother”, may repeat the attack if the second stage B attack fails. Here, A + B + C means the complete autonomous spacecraft during the approach phase to NEA.

**Figure 3 materials-15-01752-f003:**
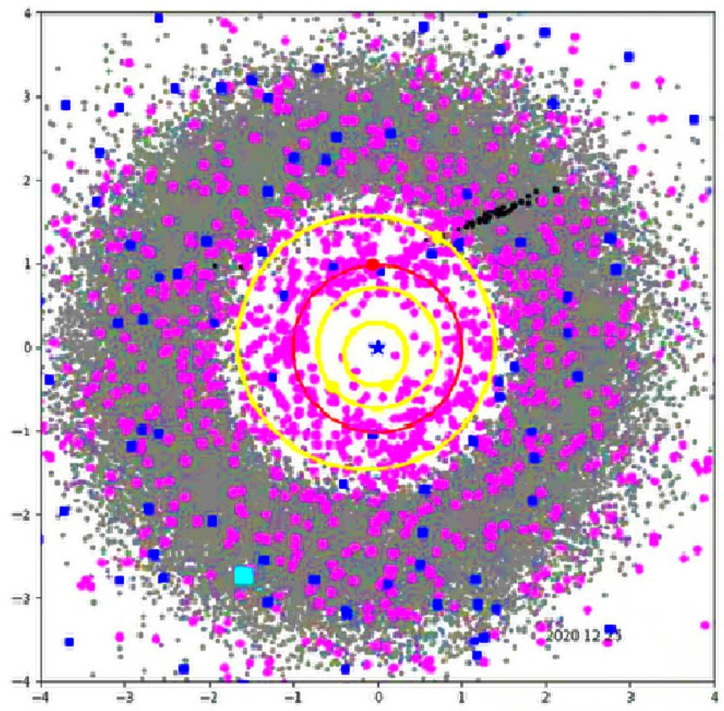
Top–down view of the solar system showing the position, on 25 December 2020, of all the Main Belt asteroids (grey circles), near-Earth asteroids (pink circles), and comets (blue squares) detected by the near-Earth objects Wide-field Infrared Survey Explorer (NEOWISE) during the first five years of the reactivation survey. The black circles show the objects detected during the final week of year five. The yellow circles and points indicate the orbits and locations of Mercury, Venus, and Mars. Earth and its orbit are in red [15].

**Figure 4 materials-15-01752-f004:**
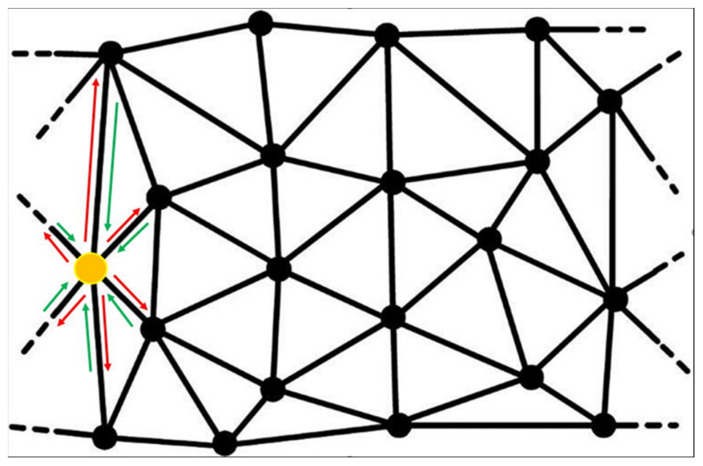
The projection of a 3D distributed network on a plane. The yellow node represents the primary spacecraft. The red arrow is the synchronization. The green arrow is the coordination.

**Figure 5 materials-15-01752-f005:**
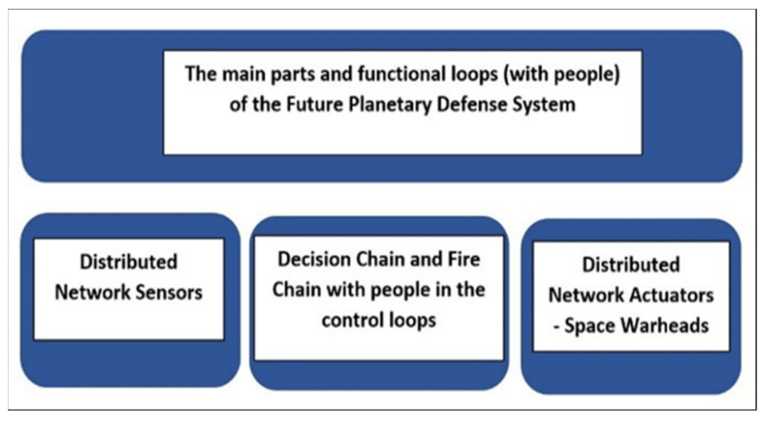
The main parts and functional loops of the Future Planetary Defense System with people in the control loops.

**Figure 6 materials-15-01752-f006:**
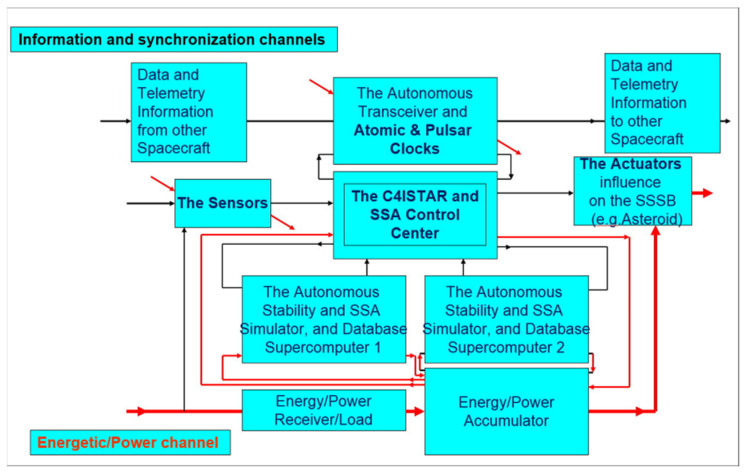
The modification and extension of Marian Mazur’s scheme of an autonomous system with application to FPDS’s spacecraft. SSSB—small solar system body; SSA—space situational awareness.

**Figure 7 materials-15-01752-f007:**
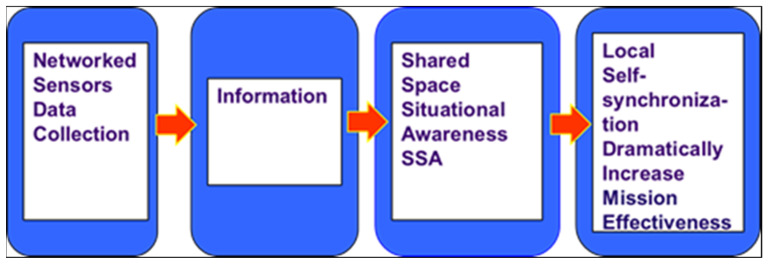
Slightly modified Network-Centric Operations and Warfare (NCW) tenets.

**Figure 8 materials-15-01752-f008:**
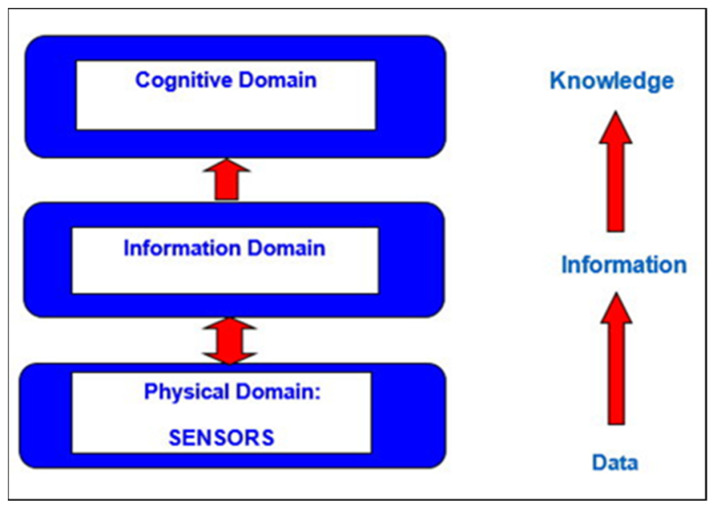
Knowledge-based approach.

**Figure 9 materials-15-01752-f009:**
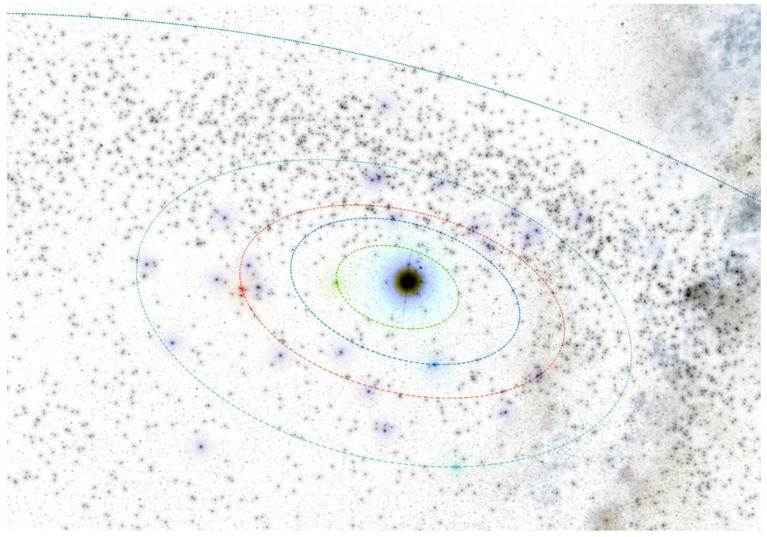
Snapshot from a 3D interactive asteroid space visualization—Asterank [32]. Details on orbits and basic physical parameters are from the Minor Planet Center and NASA JPL. In the center—the Sun, Green—Mercury orbit, Blue—Venus orbit, Red—Earth orbit, Cyan—Mars orbit, Dark blue—Jupiter orbit, Main Belt lies between Mars orbit and Jupiter orbit, in which white points are asteroids.

**Figure 10 materials-15-01752-f010:**
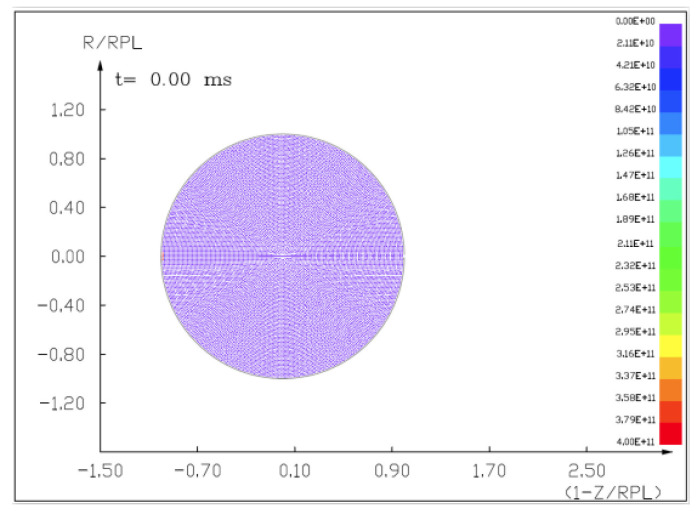
A homogeneous, computational mesh of the asteroid at the initial moment (t=0.0 s) shows the uniform pressures inside.

**Figure 11 materials-15-01752-f011:**
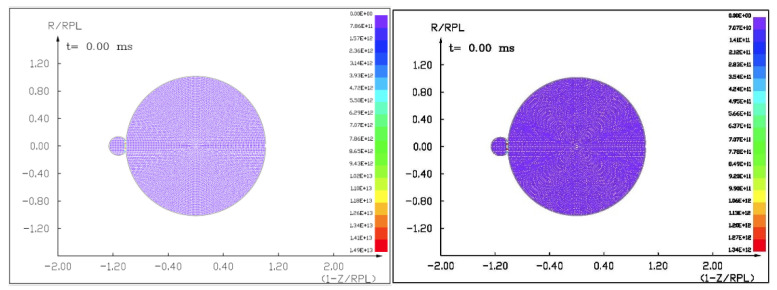
The initial stage of the computer simulation of the deflection of a 100 m diameter asteroid using a smaller 13.5 m diameter asteroid. Initial velocity of a smaller asteroid was 10 km/s and 3 km/s.

**Figure 12 materials-15-01752-f012:**
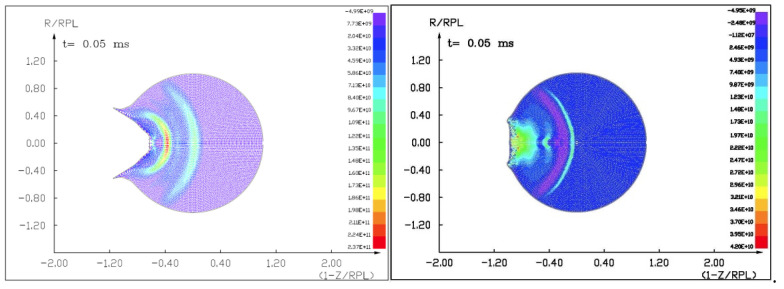
The next step in the visualization of the computer simulation of the deflection of a 100 m diameter asteroid using a smaller 13.5 m diameter asteroid. We could see a specific sequence of 2 shock waves in 5 ms after impact.

**Figure 13 materials-15-01752-f013:**
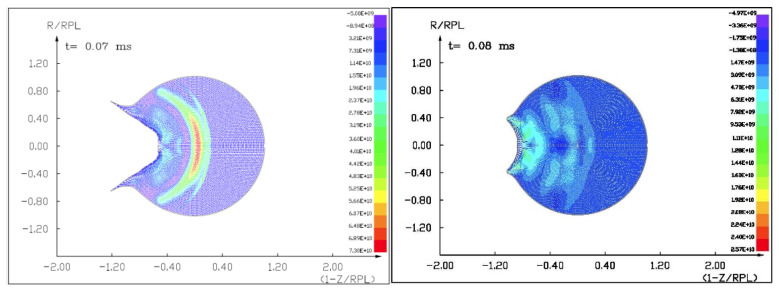
The next example of the computer simulation of the deflection of a 100 m diameter asteroid using a smaller 13.5 m diameter asteroid. We could see both shock waves moving together through a more massive asteroid in 7 ms after impact with a velocity of 10 km/s. In the next figure, 8 ms after impact with velocity 3 km/s, we could see a complicated structure of interacting shock waves.

**Figure 14 materials-15-01752-f014:**
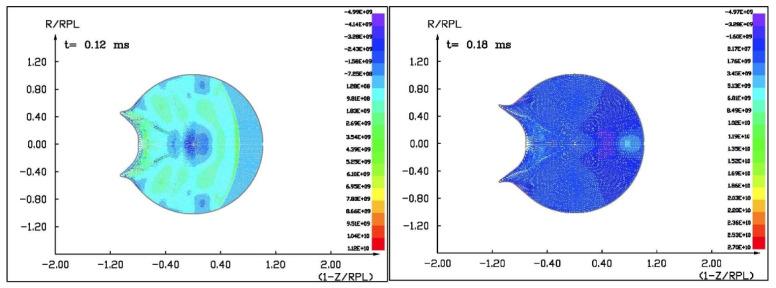
The next example of the computer simulation of the deflection of a 100 m diameter asteroid using a smaller 13.5 m diameter asteroid. In the next figure, 12 and 18 ms after impact with velocity 3 km/s, we could see a complicated structure of interacting shock waves and cratering process.

**Figure 15 materials-15-01752-f015:**
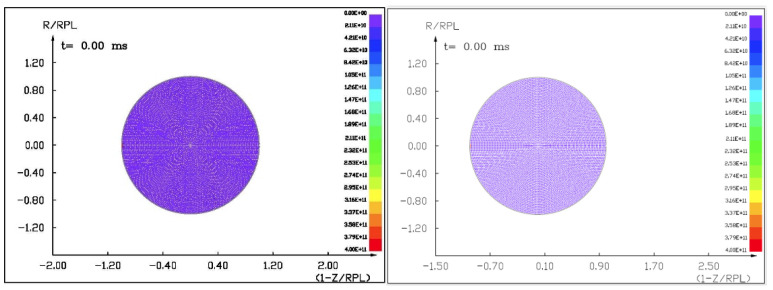
The initial stage of the computer simulation of the deflection of a 100 m diameter asteroid using conventional explosives. Initial conditions, *t* = 0, *p*_0_ = 401 kb. Octagen charge mass equals 7100 kg.

**Figure 16 materials-15-01752-f016:**
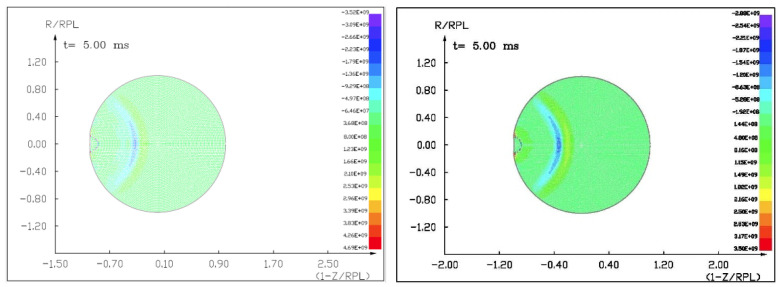
The next stage visualization of the computer simulation of the deflection of the 100 m diameter asteroid using conventional explosives. We could see the crater and front of shock in 5 ms after detonation.

**Figure 17 materials-15-01752-f017:**
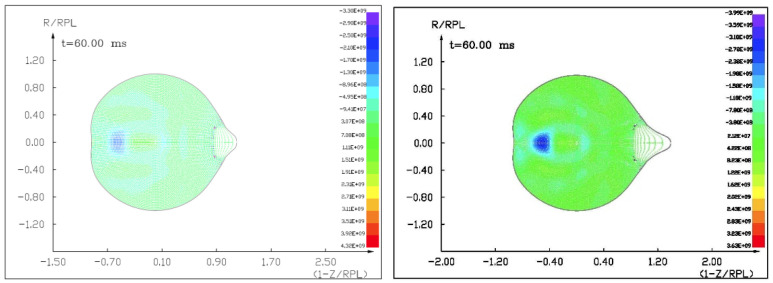
The last visualization of the computer simulation of the deflection of a 100 m diameter asteroid using conventional explosives. We could see the fragmented interior of a more massive asteroid together with the spalling phenomenon of a fragment around the antipodal point in 60 ms after detonation.

**Figure 18 materials-15-01752-f018:**
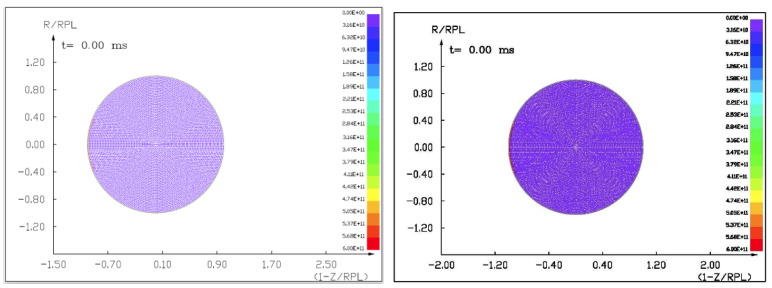
The initial stage of the computer simulation of the deflection of a 100 m diameter asteroid using a giant laser pulse. Initial conditions, *t* = 0, *p*_0_ = 600 kb. The radius of the ablation area is 20 m.

**Figure 19 materials-15-01752-f019:**
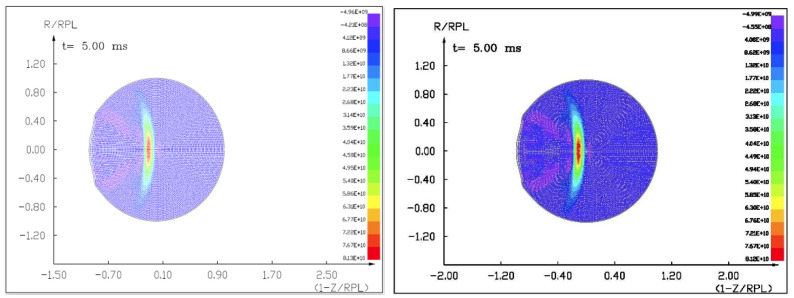
The next stage of the computer simulation of the deflection of a 100 m diameter asteroid using a giant laser pulse. We could see a strong shock wave propagation in 5 ms after ablation.

**Figure 20 materials-15-01752-f020:**
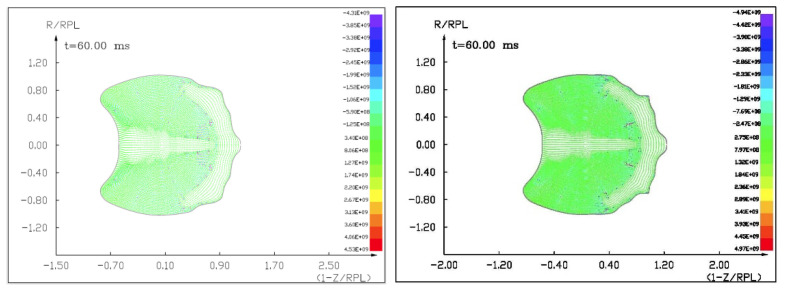
The last visualization of the computer simulation of the deflection of a 100 m diameter asteroid using a giant laser pulse. We could see the large deformation of the asteroid and completely fragmented interior together with the large spalling phenomenon of a great fragment around the antipodal point in 60 ms after ablation.

**Figure 21 materials-15-01752-f021:**
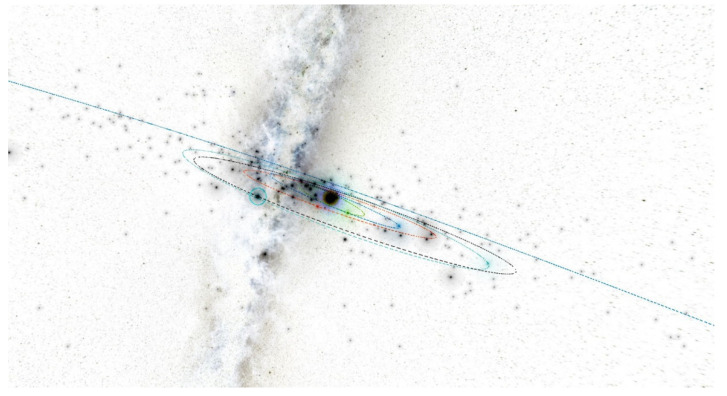
Ring of Spacecraft b0 with the ecliptic latitude b = 0° [32].

**Figure 22 materials-15-01752-f022:**
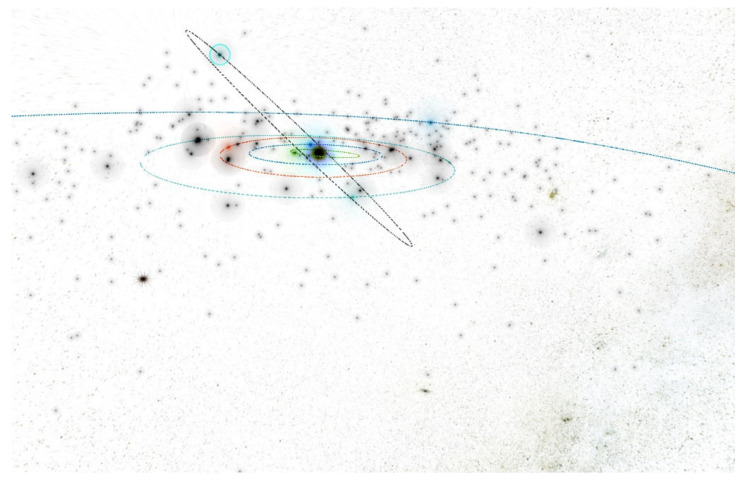
Ring of Spacecraft b45 with the ecliptic latitude b = 45° [32].

**Figure 23 materials-15-01752-f023:**
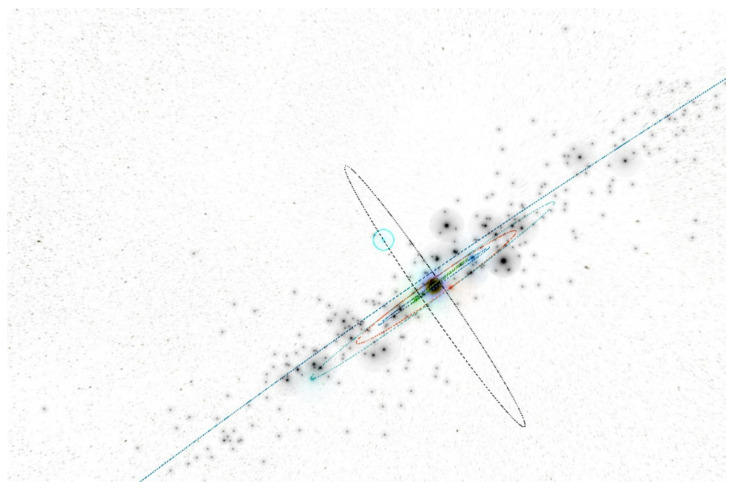
Ring of Spacecraft b90 with the ecliptic latitude b = 90° [32].

**Figure 24 materials-15-01752-f024:**
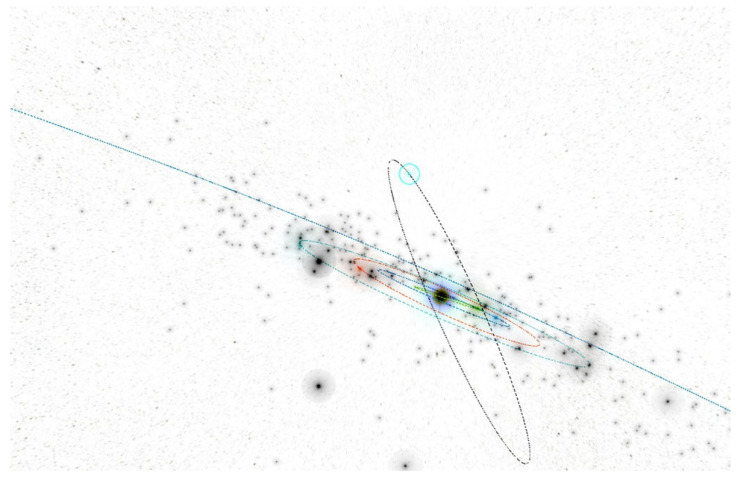
Ring of Spacecraft b-45 with the ecliptic latitude b = −45° [32].

**Table 1 materials-15-01752-t001:** Initial data and characteristics of the asteroid.

Initial Data	
Diameter	100 m;
Density of rock	2400 kg/m^3^;
Angle of impact	45 degrees;
Initial velocity	20 km/s;
Target	Sedimentary rock;
Distance to the point of impact	1 km;
Mass	1.0 × 10^10^ kg

**Table 2 materials-15-01752-t002:** Results of the calculation.

Results of Computations	
Energy before atmospheric entry	2.51 × 10^17^ Joules = 60.0 MegaTons TNT;
Impact frequency	The average interval between impacts of this size somewhere on Earth during the last 4 billion years is 4.9 · 10^3^ years.
Atmospheric entry	The projectile begins to break up at an altitude of 63,300 m. The projectile reaches the ground in a broken condition. The mass of the projectile strikes the surface at a velocity of 4.45 km/s.
The impact energy	1.24 × 10^16^ Joules = 2.97 MegaTons TNT.
The broken projectile fragments strike the ground in an ellipse of dimension	0.917 km by 0.648 km.
Crater dimensions	Crater shape is normal in spite of atmospheric crushing; fragments are not significantly dispersed.
Transient crater diameter	904 m
Transient crater depth	320 m
Final crater diameter	1.13 km
Final crater depth	240 m
The crater formed is a simple crater	
The floor of the crater is underlain by a lens of broken rock debris (breccia) with a maximum thickness of 111 m. At this impact velocity (<12 km/s), little shock melting of the target occurs.	
Airblast	The air blast will arrive approximately 3.03 s after impact.
Peak overpressure	1,580,000 Pa = 15.8 bars.
Max wind velocity	976 m/s.
Sound intensity	124 dB (Dangerously Loud).
Damage description	Multistory wall-bearing buildings will collapse.
Wood frame buildings will almost completely collapse.	
Multistory steel-framed office-type buildings will suffer extreme frame distortion, incipient collapse.	
Highway truss bridges will collapse.	
Highway girder bridges will collapse.	
Glass windows will shatter.	
Cars and trucks will be largely displaced and grossly distorted and will require rebuilding before use.	
Up to 90 percent of trees will be blown down, the remainder stripped of branches and leaves.	

## Data Availability

Data available in a publicly accessible repository.

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
