# Peer review of "Hydrocode Investigations of Terminal Astroballistics Problems during the Hypothetical Future Planetary Defense System’s Space Mission"

_materials, 2022, doi:10.3390/ma15051752_

Round 1

Reviewer 1 Report

The article is devoted to the preliminary concept of the Future Planetary
Defense System , emphasizing Astroballistics. This paper is intended to
support international efforts to improve the planetary security of the Earth.
This topic is interesting. My comment is as follows:
1.The keywords are too more.
2.The main contributions of this paper shall be stated in Introduction.
3.The format of the references shall be united and some information of the
references shall be improved.
4.Some recent related works on network systems shall be added. For example,
1)https://doi.org/10.1007/s12559-020-09782-w
2)https://doi.org/10.1002/mma.7581

Author Response

Dear Reviewer,

The authors would like to thank you for extensive reviewing the manuscript. We appreciate your comments, and we prepared the corrections according to your suggestions.

REVIEWER #1 COMMENT

“The article is devoted to the preliminary concept of the Future Planetary
Defense System , emphasizing Astroballistics. This paper is intended to
support international efforts to improve the planetary security of the Earth.
This topic is interesting. My comment is as follows:

1.The keywords are too more.”

            Dear Reviewer,

Thank you very much for your kind comments and suggestions to improve our contributions.
            We decided to reduce the number of words and shorten longer phrases:

Keywords: astronautics; astrodynamics; terminal astroballistics; hydrocode; modeling and simulation; asteroid; system architecture; space mission; navigation; planetary defense;

2.The main contributions of this paper shall be stated in Introduction.

            Dear Reviewer,

Thank you very much, according to your suggestions, we've moved our main contributions to the introduction.

3.The format of the references shall be united and some information of the references shall be improved.

Dear Reviewer,

Thank you very much for your comment and according to your suggestions, the format of the references has been harmonized and some information in the references has been corrected and will be formatted using the English editing process.

4.Some recent related works on network systems shall be added.

Dear Reviewer,

            The authors appreciate a lot your suggestions on the manuscript. At “References” we include suggested references. Thank you very much.

Reviewer 2 Report

The work is quite interesting and multidisciplinary. Work is interesting in terms of publication and further discussion.
Remarks
1. In the work, the text and drawings require formatting and bringing to a single style. Especially Pages 6, 14, 17, 18
2. The article has various formulas, as well as references to various software codes, which are argued that they are verified. However, the given text does not make up a clear and understandable picture of the used mathematical model. In this regard, the authors need to rewrite the text.

Author Response

Dear Reviewer,

 We appreciate your comments and remarks, and we prepared the corrections according to your suggestions.

REVIEWER #2 COMMENT

“The work is quite interesting and multidisciplinary. Work is interesting in terms of publication and further discussion.

Remarks

  1. In the work, the text and drawings require formatting and bringing to a single style. Especially Pages 6, 14, 17, 18”

            Dear Reviewer,

Thank you very much for your kind comments and remarks with your suggestions. We formatted the text and drawings that required it, especially pages 6, 14, 17, 18, to achieve a uniform style and will be formatted using the English editing and layout process

  1. The article has various formulas, as well as references to various software codes, which are argued that they are verified. However, the given text does not make up a clear and understandable picture of the used mathematical model. In this regard, the authors need to rewrite the text.

            Dear Reviewer,

After your suggestion, we would like to explain that the applied mathematical-physical model of mechanics of a continuous medium with the equation of the state (EOS)  of a solid was presented in a form convenient for numerical implementation of Generalized Finite Difference Method (GFD). We have included in this article the rationale for verifying the code. The transcribed text is explained in summary.  At “References” we include suggested references. Thank you very much.

Reviewer 3 Report

The article is interesting but needs to be corrected.

The list ref. needs to be completed (need to add references ):

10.1016/0083-6656(56)90021-6

10.1177/0142331220987917

10.1016/j.asr.2021.02.034

Abstract:  - this section should be halved and only the facts about the publication should be written. Lots of common words.

Figure 3, 6,9,10,14 quality needs to be improved. The drawings are of poor quality. I ask you to make more clear explanations on these figures.

Recheck formulas and special characters in them. Perhaps you can reduce the number of formulas and optimize the calculations. The presented models are very confusing. Explanations are bad.

The Conclusions section needs to be optimized, the authors have written too much. They do not adhere to European standards for the quality of writing publications.

Author Response

Dear Reviewer,

 The authors appreciate a lot your comments on the manuscript, and we prepared the corrections according to your suggestions.

REVIEWER #3 COMMENT

The article is interesting but needs to be corrected.

Dear Reviewer,

Thank you very much for your comments and ideas to improve our contribution.

Remarks

Abstract:  - this section should be halved and only the facts about the publication should be written. Lots of common words.

Dear Reviewer,

Following your recommendation, we have deleted half of the section  Summary, and we have left only the facts about the publication. Thank you very much.

Figure 3, 6,9,10,14 quality needs to be improved. The drawings are of poor quality. I ask you to make more clear explanations on these figures

Dear Reviewer,

These drawings have been improved and presented as negatives to enhancing their quality. Thank you for your remarks.

Recheck formulas and special characters in them. Perhaps you can reduce the number of formulas and optimize the calculations. The presented models are very confusing. Explanations are bad.

Dear Reviewer,

Thank you very much for your comments and ideas to improve our contribution. We checked and rewritten the formulas and corrected the special characters in them. The presented models look confusing, but they are used to make numerical calculations with high accuracy. We tried to improve the explanations.

The Conclusions section needs to be optimized, the authors have written too much. They do not adhere to European standards for the quality of writing publications.

            Dear Reviewer,

The conclusion section has been rewritten, shortened, and optimized. Thanks to the comments, the authors try to bring them closer to European standards of the quality of writing publications. At “References” we include suggested references. Thank you very much.
